# High Sensitivity Cryogenic Temperature Sensors Based on Arc-Induced Long-Period Fiber Gratings

**DOI:** 10.3390/s22197119

**Published:** 2022-09-20

**Authors:** Oleg V. Ivanov, Paulo Caldas, Gaspar Rego

**Affiliations:** 1Ulyanovsk Branch of Kotel’nikov Institute of Radio Engineering and Electronics of Russian Academy of Sciences, Ulitsa Goncharova 48, 432071 Ulyanovsk, Russia; 2S.P. Kapitsa Research Institute of Technology, Ulyanovsk State University, Ulitsa L. Tolstogo 42, 432017 Ulyanovsk, Russia; 3Escola Superior de Tecnologia e Gestão, Instituto Politécnico de Viana do Castelo, 4900-347 Viana do Castelo, Portugal; 4Center for Applied Photonics, Institute for Systems and Computer Engineering, Technology and Science—INESC TEC, Rua Dr. Roberto Frias, 4200-465 Porto, Portugal

**Keywords:** cryogenic temperature, optical fiber sensor, long-period fiber grating, arc-induced grating, dispersion turning points

## Abstract

In this paper, we investigated the evolution of the dispersion curves of long-period fiber gratings (LPFGs) from room temperature down to 0 K. We considered gratings arc-induced in the SMF28 fiber and in two B/Ge co-doped fibers. Computer simulations were performed based on previously published experimental data. We found that the dispersion curves belonging to the lowest-order cladding modes are the most affected by the temperature changes, but those changes are minute when considering cladding modes with dispersion turning points (DTP) in the telecommunication windows. The temperature sensitivity is higher for gratings inscribed in the B/Ge co-doped fibers near DTP and the optimum grating period can be chosen at room temperature. A temperature sensitivity as high as −850 pm/K can be obtained in the 100–200 K temperature range, while a value of −170 pm/K is reachable at 20 K.

## 1. Introduction

Systems operating at cryogenic temperatures are becoming increasingly important in the energy sector, transportation, medicine technology and many other fields (high energy physics, military, aerospace, etc.) [1,2,3]. Some applications also use superconducting magnets and high magnetic fields requiring electromagnetic shield and heat transfer control of their monitoring devices [4,5,6]. The standard equipment used in these environments may exhibit critical drawbacks [7,8] and, therefore, researchers have been looking for the intrinsic advantages of using optical fiber sensors [9,10]. Recently, different approaches have been tested ranging from distributed sensing, based on Raman, Rayleigh and Brillouin backscattering, to the use of interferometric and other wavelength-selective devices [11,12,13]. Nevertheless, fiber Bragg gratings (FBGs) have been the most explored, despite showing low-temperature sensitivity below 100 K [14,15]. For this reason, several techniques have been applied in order to enhance their thermal sensitivity, such as the deposition of metal or sol–gel coatings on the fiber cladding or by embedding or bonding them to substrates with very different thermal expansion coefficients [16,17,18,19]. When compared to conventional FBGs, the fabrication of modified FBGs is time consuming and the temperature sensitivity improvement is limited [20]. On the other hand, it is known that long-period fiber gratings (LPFGs) possess higher temperature sensitivity, typically an order of magnitude higher for temperatures above room temperature [21]. However, research involving the use of LPFGs at cryogenic temperatures is very scarce [22,23,24], possibly due to two main factors: they need access to both sides of the fiber, and they are also very sensitive to bending. In 2015 [25], we solved the problem by using a phase-shift LPFG (PS-LPFG) working in reflection placed inside a polyamide capillary with a 1 mm inside diameter. The results obtained were considerably better than for other fiber-grating sensors, but still limited at temperatures close to 0 K. In 2016 [26], we proved that it is possible to fabricate LPFGs with grating periods as short as 148 μm using the electric arc discharge technique. This allows the fabrication of LPFGs in the dispersion turning points, the region where they exhibit the highest sensitivity to changes in physical parameters such as temperature. This paper investigates, for the first time, to the best of our knowledge, the temperature dependence of the dispersion curves in order to find the optimum feasible grating period that allows the increase of the temperature sensitivity at cryogenic temperatures. For that goal, we performed computer simulations based on the experimental data that we obtained for PS-LPFGs inscribed in the SMF-28 Corning fiber and also on the B/Ge co-doped PS1250/1500 Fibercore fiber.

## 2. Long-Period Fiber Grating Structure

LPFGs are periodic structures with grating periods ranging from 100 μm up to 1 mm. Their transmission spectra exhibit a number of dips at specific wavelengths that satisfy the resonance condition, corresponding to coupling of the core mode to different co-propagating cladding modes [27]. In this paper, gratings were arc-induced in a standard single-mode telecommunication GeO_2_-doped silica fiber, the SMF-28 fiber from Corning, and in two batches of a photosensitive fiber from Fibercore, the PS 1250/1500 GeO_2_-B_2_O_3_ co-doped silica fiber. The first, with a core radius of 4.3 μm and a cut-off wavelength of 1.31 μm [28], results in a refractive index difference of ~5 × 10^−3^. The two Fibercore fibers have the following parameters: B/Ge#1 (NA = 0.13, MFD = 9.6 μm, λ_cut-off_ = 1.15 μm) and B/Ge#2 (NA = 0.14, MFD = 8.9 μm, λ_cut-off_ = 1.24 μm), leading to a core radius of ~3.4 μm and a refractive index difference of ~6 × 10^−3^ and ~7 × 10^−3^, respectively.

The gratings were arc-induced [29] in those fibers using two different high-voltage power supplies: a homemade one [26] (Figure 1) that enables the writing of LPFGs in the dispersion turning points (DTP), and a commercial fusion splice machine (BICC AFS130) [30] used typically to fabricate gratings above 400 μm. The gratings used in 2015 for cryogenic temperature measurements [25] were produced in the SMF-28 fiber and in the B/Ge #1 fiber using the fabrication parameters: grating period Λ, electric current I, number of arc discharges N, arc duration Δt and pulling weight w, presented in Table 1.

On the other hand, to achieve gratings in the DTP, we used the homemade high-voltage power supply and the fabrication parameters presented in Table 2 [26].

It should be stressed that under our fabrication conditions, coupling occurs to LP_1*j*_ cladding modes in the SMF-28 fiber and to LP_0*j*_ cladding modes in the B/Ge co-doped fibers [31,32,33]. Moreover, depending on the fabrication conditions, a geometric modulation of the fiber cross section may occur, which is particularly noticed for high pulling weight [34]. It is also accompanied with the creation of new stresses in the fiber [35]. On the other hand, for low pulling weight, the annealing of intrinsic stresses/viscoelastic stresses is observed, which in turn leads to an increase of the cladding refractive index of the order of 1 × 10^−4^ [36]. This value is compatible with the fact that, for the same grating period, the resonances of arc-induced gratings in the SMF-28 fiber are shorter by ~50 nm (depending on the cladding modes) than the ones belonging to mechanically induced gratings [37]. A small decrease of the core refractive index may also occur [38,39]. Furthermore, the dimensions of the arc-discharge, its electric current and duration will impact the length of the region affected by the arc. Thus, the duty cycle of the index modulation will decrease as the grating period increases. Therefore, the effects of the arc-discharge need to be considered in the computer simulations.

## 3. Refractive Index of the Fiber Structure

For calculating the spectra of a LPFG, we need to define the refractive indices of the fiber structure: first, the core and cladding refractive indices for the step index fiber or the continuous distribution of the refractive index along the radial direction for fibers with gradient profiles; second, the modulation of the refractive index along the fiber that creates the grating. In contrast to a fiber Bragg grating, precise knowledge of the difference between the core and cladding refractive indices is required due to the fact that the resonance condition for the LPFG involves a difference between the propagation constants of the core (*β*_co_) and cladding modes (*β*_cl_):(1)$$\beta_{\mathrm{co}}(\lambda_{r})-\beta_{\mathrm{cl}}(\lambda_{r})=\frac{2\pi }{\Lambda },$$

It may be difficult to know the refractive indices, since their difference is determined by the concentration of various dopants having different wavelength dispersion. The material dispersion of doped silica is also important, because the LPFG spectrum covers a broad range of wavelengths. 

Thus, we should construct the dependence of the refractive index of silica on four parameters: wavelength, germanium concentration, boron concentration, and temperature. The ranges are 0.9–1.7 μm for wavelength, 0–10% for germanium concentration, 0–20% for boron oxide concentration, and 0–300 K for temperature. This dependence is a four-dimensional function. There is limited information on this function in the literature, which provides different slices for some parameters to be fixed. For example, the dispersion of germanium- and boron/germanium-doped silica is known only at room temperature [40,41], but the thermo-optic coefficient is only measured for pure silica [42], etc. 

In general, as far as fused silica is concerned, the Sellmeier equation [43] is used, which empirically relates the refractive indices as a function of wavelength:(2)$$n{\left(\lambda \right)}^{2}=1+\sum_{j=1}^{3}\frac{D_{S,j}\lambda^{2}}{\lambda^{2}-\lambda_{S,j}^{2}},$$

In this equation, the Sellmeier coefficients (*D*_s,*j*_ and *λ*_s,*j*_) found in [43] describe the refractive index of pure fused silica at room temperature. The fiber core doped with various dopants has a modified refractive index. In this paper, we consider germanium- and boron/germanium-doped fibers. For these types of doping, we can use Sellmeier’s coefficients found by Fleming [40,41]: (3)$$n{\left(\lambda \right)}^{2}=1+\sum_{j=1}^{3}\frac{\left(c_{S}D_{S,j}+c_{G}D_{G,j}+c_{B}{D^{\prime }}_{B,j}\right)\lambda^{2}}{\lambda^{2}-{\left(c_{S}\lambda_{S,j}+c_{G}\lambda_{G,j}+c_{B}{\lambda^{\prime }}_{B,j}\right)}^{2}},$$

In Equation (3), we used corrected coefficients for boron oxide that were obtained from ${D^{\prime }}_{B,j}=D_{G,j}+\left(D_{B,j}-D_{G,j}\right)/0.133$, and ${\lambda^{\prime }}_{B,j}=\lambda_{G,j}+\left(\lambda_{B,j}-\lambda_{G,j}\right)/0.133$, since the initial coefficients $D_{B,j}$ were obtained for boron-doped silica with a concentration of 13.3%. So, if we calculate the refractive index of silica with a boron oxide concentration of 13.3%, we obtain exactly the refractive index measured. The numerical values of the coefficients are given in Table 3. The indices S, G and B in the coefficients stand for silica, germanium and boron.

We assume that the changes in the refractive index induced by temperature are small; therefore, we can employ an approximation: (4)$$n(\lambda ,c_{\mathrm{Ge}},c_{B},T)=n(\lambda ,c_{\mathrm{Ge}},c_{B},T_{0})+\int_{T_{0}}^{T}\eta (c_{\mathrm{Ge}},c_{B},T)dT,$$
where $T_{0}$ is the room temperature, η=dn/dT is the thermo-optic coefficient of silica, and $c_{\mathrm{Ge}},c_{B}$ are the concentrations of germanium and boron dopants. The thermo-optic coefficient can be, in principle, wavelength dependent; however, experimental measurements of the dispersion of pure silica as a function of temperature presented in [42] have shown that this dependence is weak and can be neglected (measurements show that the value is ~8.2 × 10^−6^ in the third telecommunication window and the relative difference is within 2% from 1.0 μm up to 1.7 μm). 

In order to determine the thermo-optic coefficient of pure silica, we have used Sellmeier’s equation with temperature-dependent coefficients to generate the wavelength-dependent refractive indices valid for wavelengths between 0.4 μm and 2.6 μm and for the temperature range from 30 K up to 310 K [42]. To assess the results, we calculated the Sellmeier’s coefficients for room temperature (295 K) and we compared the refractive index values with the ones obtained by Fleming [44] for quenched SiO_2_ fibers, and the difference is better than 6 × 10^−5^ for the wavelength range from 1.0 μm up to 1.7 μm. Note that the values achieved are slightly higher (<4 × 10^−4^) than those obtained using the coefficients presented by Malitson [43]. After validation, we fitted the refractive index as a function of temperature by a 6th order polynomial for a wavelength of 1.55 μm: (5)$$n(1.55,0,0,T)=\sum_{i=1}^{6}A_{i}T^{i},$$
$$\begin{array}{l} A_{0}=1.4430840,\;A_{2}=1.72833\cdot 10^{-8},A_{3}=-1.14742\cdot 10^{-12},\\ A_{4}=-9.25255\cdot 10^{-15},A_{5}=-5.14342\cdot 10^{-17},A_{6}=8.70482\cdot 10^{-20}. \end{array}$$
where the temperature is measured in Kelvin. There is no linear term in this sum, because it is known that the thermo-optic coefficient goes to zero as the temperature approaches 0 K [45,46,47]. The curve has a parabola shape at 0 K. This is experimentally corroborated by the fact that gratings are essentially insensitive to temperature changes near 0 K [15,23,25,48]. Thus, we extrapolated the refractive indices down to 0 K and the results show that the equation obtained by Leviton and Frey may be applied to lower temperatures.

Through derivatives, we obtained the thermo-optic coefficient: (6)$$\frac{dn}{dT}=\sum_{i=2}^{6}iA_{i}T^{i-1}$$

Figure 2 demonstrates the refractive index change and the thermo-optic coefficient of pure silica for a wavelength of 1.55 μm as a function of temperature.

The problem of calculating the LPFG spectra is that the wavelength shift of the LPFG resonances with temperature is determined by the difference between the thermo-optic coefficients of pure and doped silica of the cladding and the core, respectively. We may reconstruct this dependence from cryogenic experiments with fiber Bragg gratings induced in germanium and boron/germanium fibers [49]. However, not only is the range of temperatures studied in that work above 77 K, but important parameters related to the fibers used and also to the gratings inscribed are missing and, therefore, the accuracy of the results would be insufficient.

It is well known that, at room temperature, the thermal expansion coefficient can be neglected in comparison to the thermo-optic coefficients. However, at low temperatures the thermo-optic coefficients and the thermal expansion coefficients are of the same order of magnitude, and the latter may even become negative [45,50]. The thermal expansion coefficient depends on the fiber composition and also on its thermal history, that is, it depends on its fictive temperature, which is different for the core and cladding region [51].

For the SMF-28 fiber, a germanium-doped fiber, by knowing the NA, we used the additive model [52,53,54] to determine the concentration of GeO_2_ to be ~3.2 mol%. For pure GeO_2_ glass, the physical parameters used are from ref. [55]. Afterwards, we used the Sellmeier’s equation for a binary glass with coefficients obtained by Fleming [40] for pure GeO_2_ glass and the ones obtained in [42] for pure silica glass. The results were also confirmed by applying the same model to the calculus of the refractive index of binary silica-based glasses under different concentration values. 

In this work, we used the results on wavelength shifts for LPFGs in germanium- and boron/germanium-doped fibers [25] to obtain the difference between the thermo-optic coefficients of pure and doped silica. Figure 3 shows the refractive index difference between the doped and pure silica as a function of temperature for SMF-28 (Corning) and B/Ge-doped fiber (Fibercore). Following the same reasoning as for the refractive index dependence, there is no linear term in the polynomial expansion on temperature:(7)$$\Delta n=\sum B_{i}T^{i},$$
where the parameters have the following values for the two types of fibers:

SMF-28:$$B_{0}=-2.13183\cdot 10^{-5},\;B_{2}=4.00467\cdot 10^{-10},B_{3}=-5.45325\cdot 10^{-13},$$

B/Ge-doped:$$\begin{array}{l} B_{0}=6.33968\cdot 10^{-4},\;B_{2}=-1.66941\cdot 10^{-8},B_{3}=-2.52548\cdot 10^{-11},\\ B_{4}=3.75429\cdot 10^{-13},B_{5}=-6.13413\cdot 10^{-16}, \end{array}$$

One more factor that influences the refractive index change of an optical fiber with temperature is thermal expansion. For the temperature dependence of the thermal expansion coefficient of pure silica, inside their validity intervals, the expression given by Okaji et al. [50] yields a value of 5.0 × 10^−7^ K^−1^ at room temperature. Using the additive model [56], we can estimate the thermal expansion coefficient for the core of the SMF-28 fiber leading to a value of 8.9 × 10^−7^ K^−1^. 

The core and the cladding have different thermal expansion coefficients, and they are deformed inhomogeneously with temperature. The solution of the problem is known, but it is too cumbersome, its effect is small, and the thermal expansion coefficients of doped silica are unknown. Moreover, when we use the refractive index difference shown in Figure 3, we already take the thermal expansion effect on the refractive index into account. Therefore, we do not specifically consider thermal expansion in our calculation of refractive index. The temperature behavior of FBGs and LPFGs can be used in future studies to estimate the thermal expansion coefficient of the fiber core and cladding.

## 4. Dispersion Curves of LPFGs

We start our analysis of LPFGs by calculating the dispersion curves of SMF-28 and B/Ge fibers at room temperature. The structures of both fibers are assumed to have a step-index profile with several concentric silica layers containing different doping. The refractive index of each layer is calculated using the procedure described in the previous section.

During the fabrication of LPFGs by arc discharges, the structure of the pristine fiber is changed: the cladding and core diameters are reduced and their refractive indices are changed due to the modification of the internal stress distribution. Therefore, we adjusted some of the fiber parameters to obtain the best fitting of simulated dispersion curves to the experimentally measured curves. 

In order to follow the procedure regarding the simulations, Figure 4 presents a schematic diagram showing all of the steps performed. 

The following parameters for the SMF-28 fiber have been found to produce the best fit to the experimental data: $r_{\mathrm{co}}=4.3$ μm, $r_{\mathrm{cl}}=62.5$ μm, and $c_{\mathrm{Ge}}=3.2\%$. The experimental data points were taken from previous studies: the blue dots (s1) belong to gratings fabricated using the AC high-voltage power supply [26] and the red triangles (s2) using the BICC AFS130 fusion splicer [30]. The simulation results are shown in Figure 5 by solid lines. The resonance wavelengths of the LPFGs are shown depending on the grating period for LP_1*j*_ cladding modes. 

In general, the resonance wavelengths increase with the grating period, while the slope of the curve grows with the mode number. For mode numbers greater than 10, the curve becomes close to a vertical line with some curvature resulting in a two-valued function of wavelength on the period. The apex of the curve is the so-called “turning point”. The first turning point appears for an LPFG with a period of 225 μm at 1.46 μm for the LP_1,10_ mode. The next modes have turning points at lower wavelengths. 

The LP_11_ mode is a special case. The dispersion curve of this mode bends in the opposite direction for a period of 470 μm at 1.05 μm. This happens because the fiber becomes a two-mode waveguide, and the LP_11_ mode is transferred to the core. When we move to shorter wavelengths, the period of the electromagnetic field becomes shorter and one more period can fit in the core. The number of modes in a waveguide is determined by the V number: $V=2\pi r_{\mathrm{co}}NA/\lambda$. Here, NA is the numerical aperture, and $r_{\mathrm{co}}$ is the core radius. Single-mode propagation is obtained when V < 2.4. With decreasing wavelength, V becomes greater than 2.4, and the fiber can guide two modes. Thus, the LP_11_ mode is transferred to the core at 1.05 μm. In fact, all cladding modes have the same behavior, but at much shorter periods and wavelengths, when they become core modes.

Boron–germanium-doped fiber is another type of fiber that is used for the inscription of LPFGs. Arc-induced gratings in this fiber produce prevailing symmetric perturbations in the core and excite the LP_0*j*_ cladding modes [31]. These modes have somewhat different dispersion curves due to other mode symmetry and significantly different parameters of the B/Ge fiber itself. The dispersion curves for LP_0*j*_ cladding modes of two B/Ge fibers are demonstrated in Figure 6: (a) B/Ge#1 and (b) B/Ge#2. The following parameters for the fiber were used to obtain the best fit of the experimental data: B/Ge#1—$r_{\mathrm{co}}=3.8$ μm, $r_{\mathrm{cl}}=64.6$ μm, $c_{\mathrm{Ge}}=8.37\%$, $c_{B}=18.9\%$; B/Ge#2—$r_{\mathrm{co}}=2.9$ μm, $r_{\mathrm{cl}}=62.5$ μm, $c_{\mathrm{Ge}}=9.9\%$, and $c_{B}=18.9\%$. The experimental data points were taken from previous studies [26]. The dispersion curves for the B/Ge#1 fiber lie lower than for the B/Ge#2 fiber, and resonances of modes LP_0*j*_ with j≥6 were experimentally measured, while modes with j starting from 1 are seen for the B/Ge#2 fiber. Three modes with j=11,12,13 have dispersion turning points in the presented range for B/Ge#1 fiber at wavelengths 1.54, 1.46, and 1.38 μm, respectively, and one dispersion turning point for the mode with j=12 for the B/Ge#2 fiber at 1.54 μm.

## 5. Sensitivity and Resolution of LPFGs 

For optical fiber sensors based on LPFGs, it is important to determine the main parameters that describe the performance such as resolution, sensitivity, and figure of merit, which is defined as the ratio between sensitivity and FWHM [57]. These parameters depend on the grating length, grating period, wavelength, mode number, and others. Among other methods, working around the dispersion turning point is often used to increase the sensitivity of the grating sensor [58]. To evaluate temperature sensitivity and figure of merit (FOM) of LPFGs, here, we use a theoretical approach along with a simulation of gratings with particular parameters.

The temperature sensitivity of an LPFG is defined as the ratio between the wavelength shift of a grating resonance in the transmission spectrum upon a temperature increase of 1 degree. The transmission coefficient of an LPFG for one of the resonances can be written in the following form [59]:(8)$$T_{\mathrm{cl}}=\frac{\kappa^{2}}{\kappa^{2}+\delta^{2}}\sin^{2}\sqrt{\kappa^{2}+\delta^{2}}L,$$

Here, κ is the coupling constant, and L is the length of the grating. The detuning parameter δ is defined as the detuning from the center of resonance: (9)$$\delta =\frac{1}{2}\left[\beta_{\mathrm{co}}(\lambda )-\beta_{\mathrm{cl}}(\lambda )-\frac{2\pi }{\Lambda }\right],$$

The maximum loss is observed in the center of the resonance $\beta_{\mathrm{co}}(\lambda_{r})-\beta_{\mathrm{cl}}(\lambda_{r})=2\pi /\Lambda$ for κL=π/2. Therefore, the first minimum is for the detuning parameter $\delta =\delta_{s}=\sqrt{3}\pi /2L$. The corresponding wavelength of the first minimum is separated from the center by $\Delta \lambda_{s}$: $\lambda_{s}=\lambda_{r}+\Delta \lambda_{s}$. Assuming that the wavelength separation is small, we can obtain its relation with the detuning parameter at the first minimum:(10)$$\delta_{s}=\frac{1}{2}{\frac{d\left(\beta_{\mathrm{co}}(\lambda )-\beta_{\mathrm{cl}}(\lambda )\right)}{d\lambda }|}_{\lambda =\lambda_{r}}\Delta \lambda_{s}=-\pi \frac{1}{\Lambda^{2}}{\left(\frac{d\lambda }{d\Lambda }\right)}^{-1}\Delta \lambda_{s},$$
and reversely
(11)$$\Delta \lambda_{s}=-\frac{\sqrt{3}}{2}\frac{\Lambda^{2}}{L}\frac{d\lambda }{d\Lambda },$$

If we define the resonance width as the wavelength span between two minima, then it is equal to $2\Delta \lambda_{s}$ (Figure 7). The derivative dΛ/dλ is the slope of the dispersion curve Λ(λ), which depends on the wavelength and the mode number.

Let us assume that there is a change in the effective refractive index for the cladding mode due to some changes in the refractive index of the fiber $n_{\mathrm{eff}}+\Delta n_{\mathrm{eff}}$. There is a corresponding change in the propagation constant of the mode $\beta_{\mathrm{cl}}=2\pi (n_{\mathrm{eff}}+\Delta n_{\mathrm{eff}})/\lambda$. This would result in a wavelength shift of the resonance position due to an addition in the detuning parameter $\delta_{n}=\pi \Delta n/\lambda_{r}$. It can be approximated as follows:(12)$$\delta_{n}=\frac{1}{2}{\frac{d\left(\beta_{\mathrm{co}}(\lambda )-\beta_{\mathrm{cl}}(\lambda )\right)}{d\lambda }|}_{\lambda =\lambda_{r}}\Delta \lambda_{n}=-\pi \frac{1}{\Lambda^{2}}{\left(\frac{d\lambda }{d\Lambda }\right)}^{-1}\Delta \lambda_{n},$$
and reversely
(13)$$\Delta \lambda_{n}=-\frac{\Lambda^{2}}{\lambda_{r}}\frac{d\lambda }{d\Lambda }\Delta n,$$

From (11) and (13), we can obtain the ratio between the wavelength shift and the resonance width, which is the figure of merit:(14)$$\frac{\Delta \lambda_{n}}{\Delta \lambda_{s}}=\frac{2}{\sqrt{3}}\frac{\Delta nL}{\lambda_{r}},$$

As one can see, this ratio is independent of the slope of the dispersion curve dΛ/dλ. Therefore, the bandwidth is proportional to sensitivity at a fixed wavelength, and the figure of merit cannot be improved by moving close to the dispersion turning point, where dλ/dΛ has higher values. It follows from (14) that the figure of merit increases with grating length and decreases with wavelength.

However, it should be stressed that the minimum temperature change detectable at cryogenic temperatures improves if the LPFG works close to DTP, since the sensitivity can increase by an order of magnitude (due to the higher slope of the dispersion curves when compared to the one of a 540 μm grating) and the overall resolution that takes into account fluctuations in the optical power source, the minimum resolution of the OSA, and the fact that the peak detection through the fitting of a Gaussian/Lorentzian curve is not particularly impacted by a slightly wider resonance [57,58].

In order to illustrate how sensitivity and figure of merit depend on the grating period and wavelength, we demonstrate it in Figure 8 with the spectrum of LPFGs in the B/Ge#1 fiber as a function of the grating period. Each vertical line represents a spectrum in color form. One can see that the width of the spectral resonances slowly grows with mode number. The amplitudes decrease with wavelengths. The thickness of each curve increases slightly with wavelength; however, the spectral width, which is measured along the wavelength axis, increases strongly with the angle of the curve slope and is inversely proportional to its cosine. So, at the turning point, the curves have slopes that are close to vertical yielding high sensitivities, but their spectral widths are very large, so that the figure of merit is not improved.

## 6. Temperature Sensitivity of LPFGs

We use the temperature dependences of fiber refractive indices (4) to calculate how the dispersion curves are changed, the shifts of the resonance wavelengths, and the figure of merit for different cladding modes. Figure 9 demonstrates the dispersion curves for SMF-28 fiber for temperatures changing between 0 and 300 K for cladding mode numbers j=1…13. The dispersion curves belonging to the lower order cladding modes are the most affected by the temperature change, but those changes occur for the smallest slopes of the dispersion curves. Much higher slopes are observed for cladding modes with j=10...13 near the DTP.

The resonance wavelengths are blue-shifted with decreasing temperature, if the range before the dispersion turning point is considered. The dependence of the wavelength shift on temperature is shown in Figure 10 for several cladding modes. We can see that the shift is generally larger for higher-order modes due to their higher slopes of dispersion curves (except for LP_11_ mode). The shift is highest when the DPT is approached. 

As we discussed above, high sensitivity, as a rule, is compromised by low resolution, and the figure of merit defined as the relation between the wavelength shift and the resonance width is a more appropriate value to calculate. In Figure 11, we show the figure of merit for different cladding modes of the SMF-28 fiber as a function of wavelength for a grating with a length of 20 mm and temperature change ΔT=−10 K. We note that the grating period is not fixed in this figure; rather, it is adjusted to obtain a mode resonance at a certain wavelength. In general, the absolute value of the figure of merit decreases with the mode number and wavelength. A special case is the LP_11_ mode, which behaves non-monotonically and has an extremum near 1.4 μm. So, in terms of the figure of merit, it is preferable to use lower-order modes at shorter wavelengths. This is somewhat contrary to what is required for high sensitivity. 

Let us consider the behavior of the B/Ge fiber gratings at cryogenic temperatures. Figure 12 depicts series of dispersion curves for 13 modes at temperatures from 300 K to 0 K with a step of 30 K for the B/Ge#1 fiber. The resonance wavelengths are red-shifted with decreasing temperature, if the range before the dispersion turning point is considered. The curves become more condensed when approaching 0 K.

The dependence of the wavelength shift on temperature is shown in Figure 13 for the cladding modes with j=1...13, when the resonance is at 1.3 μm (corresponding to different periods). We can see that the shift is larger for higher-order modes, it monotonically grows with decreasing temperature and the sensitivity tends to zero at 0 K. These curves follow the same trend as the difference between the core-cladding refractive index difference in Figure 3. The maximum sensitivity is in the temperature range 100–200 K and is as high as −850 pm/K for cladding mode with j=13. The temperature sensitivity is −170 pm/K at 20 K for the same mode and tends to zero at 0 K, since the thermo-optic coefficient vanishes at this temperature. These values are considerably higher than those obtained in our previous work using phase-shifted LPFGs (Figure 14). Table 4 summarizes the typical values of sensitivity obtained in the temperature range of 0–300 K using fiber gratings: coated and uncoated FBGs, LPFGs and PS-LPFGs. As it can be observed, the temperature sensitivity of LPFGs is larger than that obtained for FBGs, even the coated ones, and this work reveals that the sensitivity can duplicate using LPFGs in the dispersion turning points. 

The dependence of the figure of merit for a 20 mm grating in the B/Ge#1 fiber is demonstrated in Figure 15 for modes LP_11_–LP_1,13_ and ΔT=10 K. The curves for different modes almost coincide and decrease monotonically with wavelength. Thus, in terms of the figure of merit, it is preferable to use shorter wavelengths independently of the mode number. The period should be chosen so that a resonance at a certain wavelength is obtained. As we can see by comparing this with Figure 10, the figure of merit for the B/Ge fiber is one order of magnitude higher than for the standard fiber. The figure of merit is not improved near the DTP, because the high sensitivity is compensated by a wider resonance width.

## 7. Conclusions

We studied the evolution of the dispersion curves of long-period fiber gratings from room temperature down to 0 K. We considered gratings arc-induced in the SMF-28 fiber and in two B/Ge co-doped fibers. Computer simulations were performed based on previously published experimental data. 

We demonstrated that the dispersion curves belonging to the lower-order cladding modes are the most affected by the temperature change, but the shift is generally larger for higher-order modes due to their higher slopes of dispersion curves. The shift is largest when the dispersion turning point is approached. The shift of dispersion curves for the B/Ge fibers is one order of magnitude higher than for the SMF-28 fiber.

We have shown that two parameters important for the implementation of a temperature sensor are the sensitivity (wavelength shift per unit of temperature change) and the figure of merit (wavelength shift related to resonance width). A temperature sensitivity as high as −850 pm/K can be obtained in the 100–200 K temperature range. The temperature sensitivity is −170 pm/K at 20 K and tends to zero at 0 K due to the vanishing thermo-optic coefficient. The figure of merit is higher at shorter wavelengths and is independent of the mode number.

In future work, we plan to perform cryogenic temperature measurements on LPFGs with grating periods shorter than 200 μm, inscribed in B/Ge#1 fiber, and we also intend to investigate the second resonance of the LP_11_ mode at shorter wavelengths and determine its turning point.

## Figures and Tables

**Figure 1 sensors-22-07119-f001:**
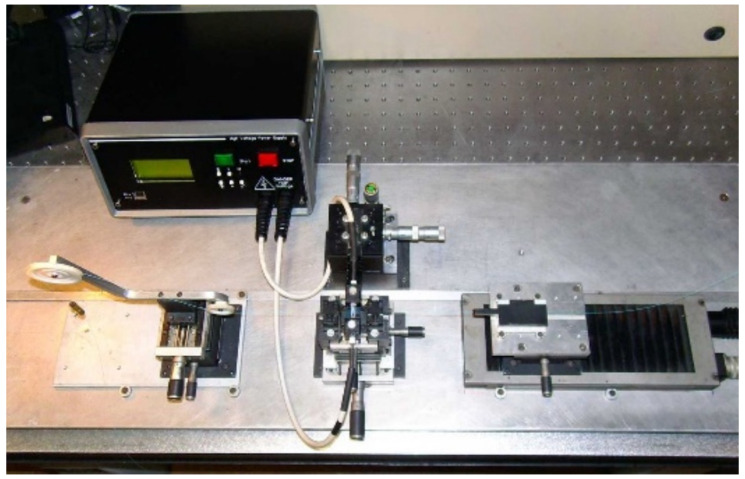
Experimental setup for LPFG fabrication in the dispersion turning points. Reprinted with permission from ref. [26]. © 2022 IEEE.

**Figure 2 sensors-22-07119-f002:**
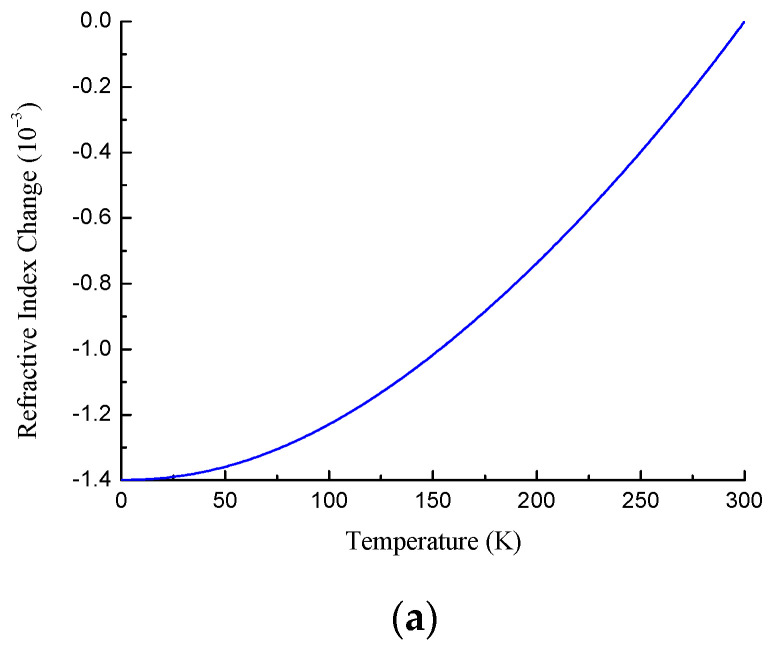
(**a**) Refractive index change and (**b**) thermo-optic coefficient of pure fused silica at 1.55 μm as a function of temperature.

**Figure 3 sensors-22-07119-f003:**
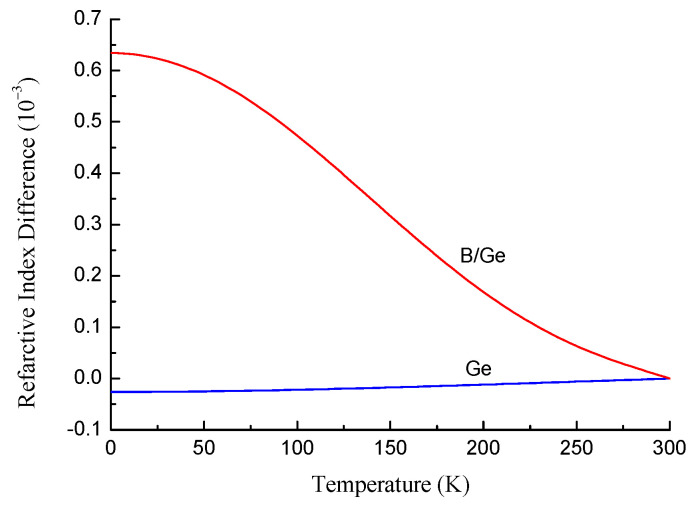
Refractive index difference between Ge- and B/Ge-doped silica and pure silica as a function of temperature.

**Figure 4 sensors-22-07119-f004:**
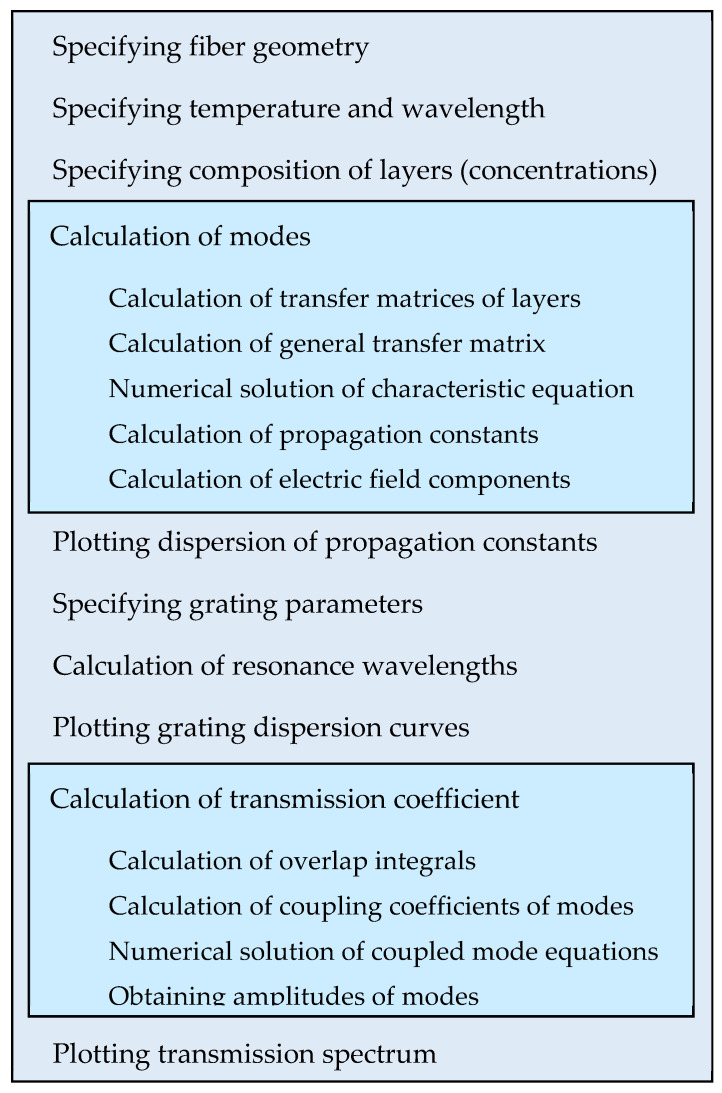
Schematic diagram of the simulations.

**Figure 5 sensors-22-07119-f005:**
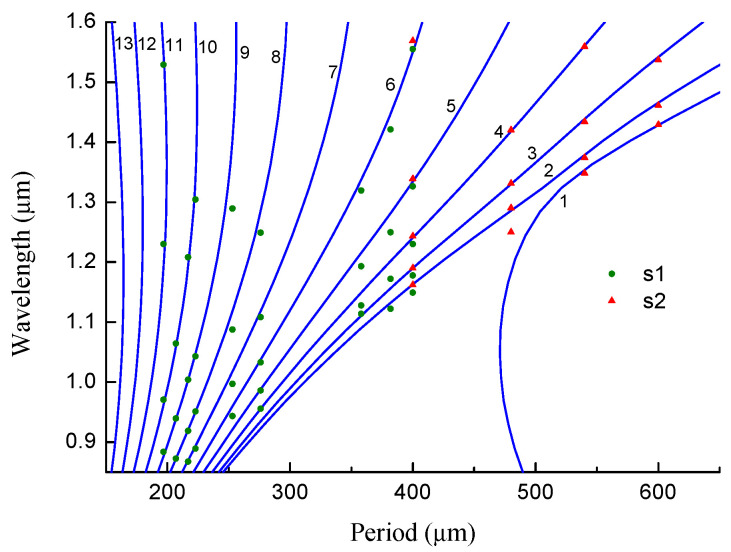
Resonance wavelengths of LPFGs vs. grating period for different LP_1*j*_ cladding modes in the SMF-28 fiber.

**Figure 6 sensors-22-07119-f006:**
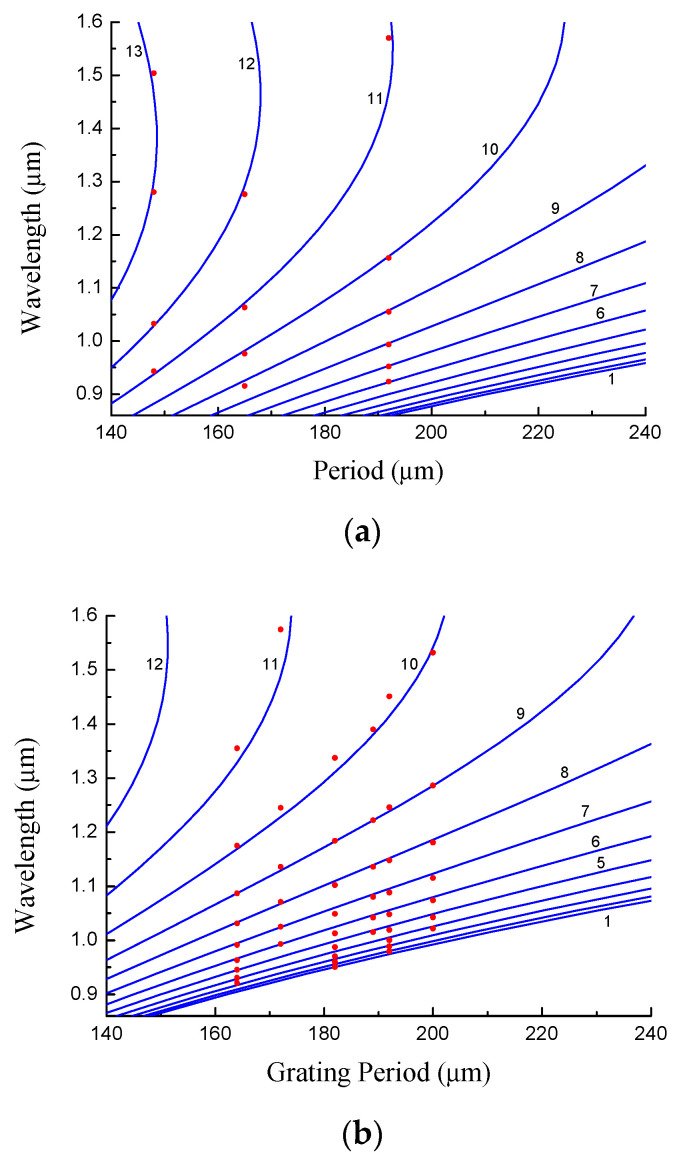
Resonance wavelengths of LPFGs vs. grating period for different LP_0*j*_ cladding modes in (**a**) B/Ge#1 and (**b**) B/Ge#2 fibers.

**Figure 7 sensors-22-07119-f007:**
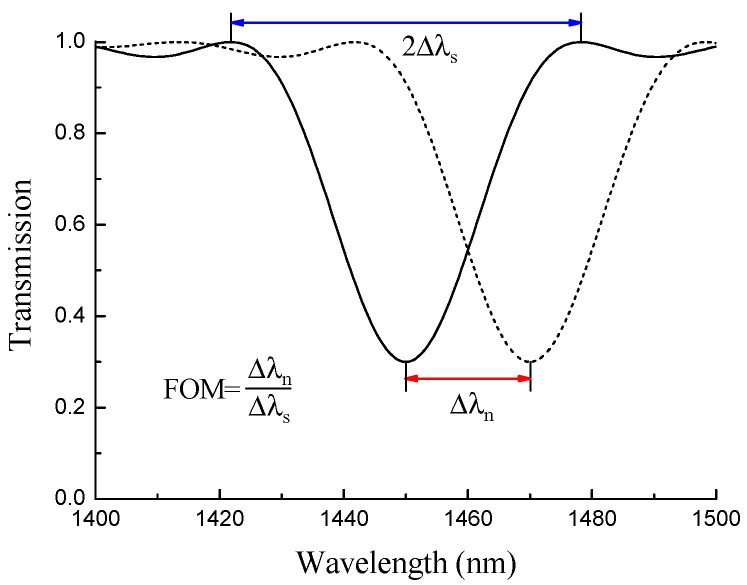
Illustration of figure of merit for a LPFG resonance shifting in wavelength.

**Figure 8 sensors-22-07119-f008:**
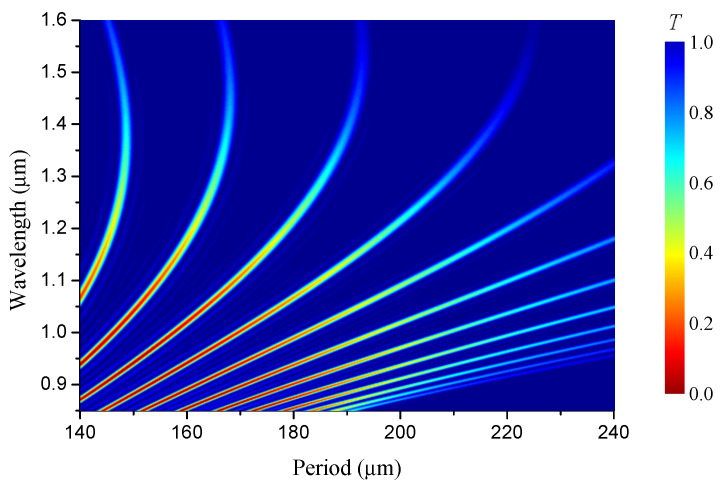
Spectra of LPFGs in B/Ge#1 fiber vs. grating period.

**Figure 9 sensors-22-07119-f009:**
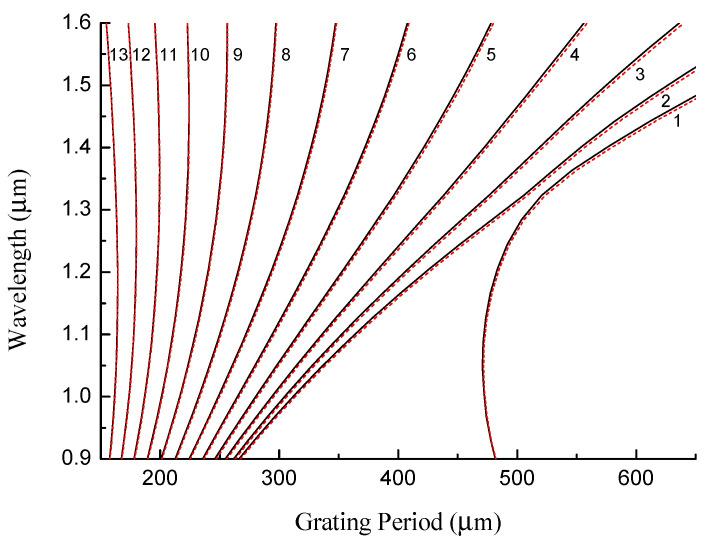
Dispersion curves for SMF-28 fiber for temperatures changing between 0 (red dashed lines) and 300 K (black solid lines) for cladding mode numbers j=1...13.

**Figure 10 sensors-22-07119-f010:**
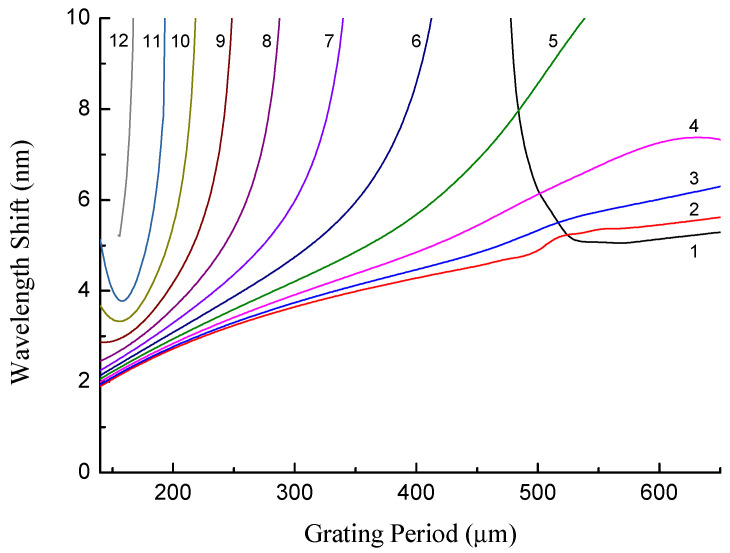
Absolute value of the wavelength shift for LP_1*j*_ modes as a function of grating period for the temperature change from 300 K to 0 K.

**Figure 11 sensors-22-07119-f011:**
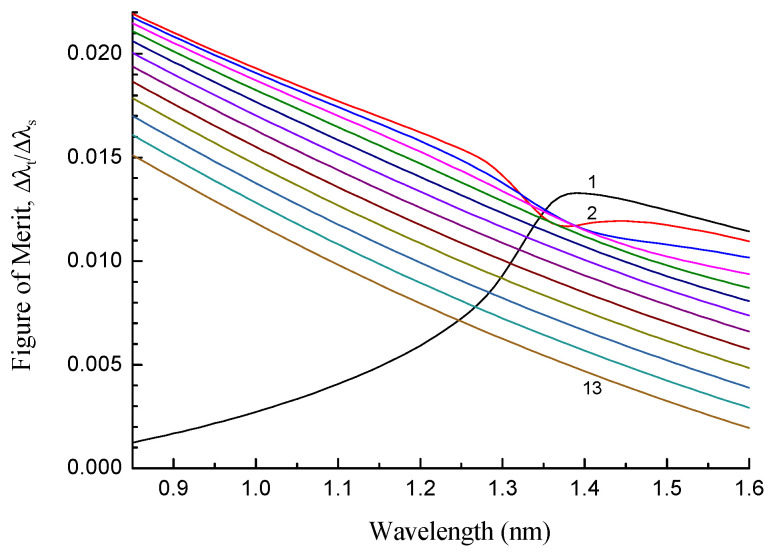
Figure of merit for different cladding modes of a grating with a length of 20 mm in SMF-28 fiber as a function of wavelength for ΔT=−10 K (300–290 K).

**Figure 12 sensors-22-07119-f012:**
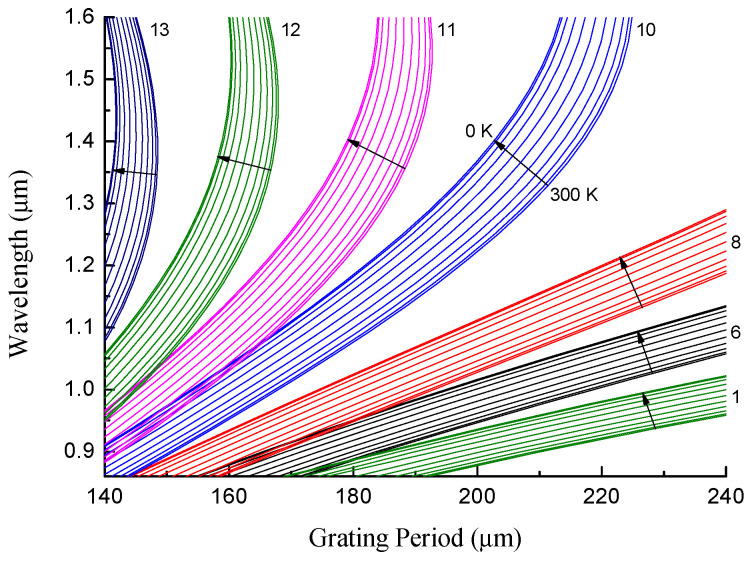
Dispersion curves for B/Ge#1 fiber for temperature changing from 300 to 0 K for cladding mode numbers j=1…13.

**Figure 13 sensors-22-07119-f013:**
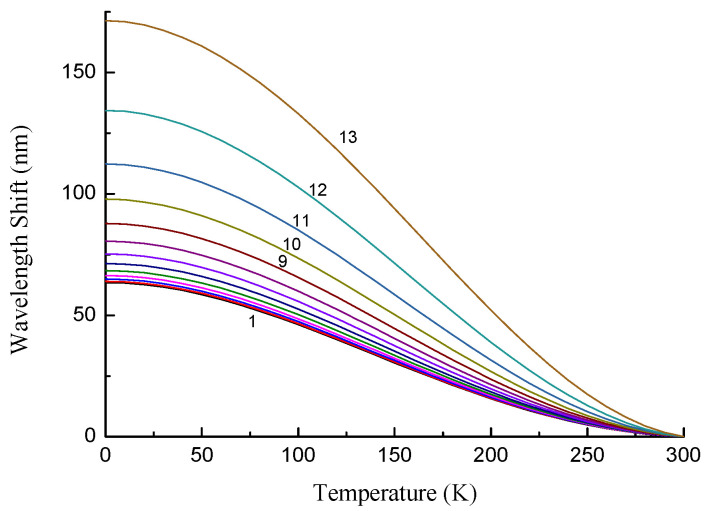
Wavelength shift in B/Ge#1 fiber for the resonances at 1.3 μm as a function of temperature.

**Figure 14 sensors-22-07119-f014:**
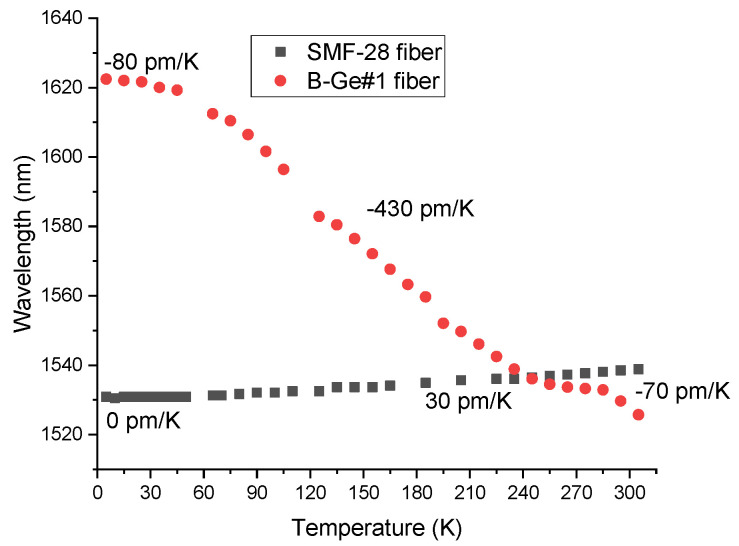
Temperature dependence of the resonance wavelengths of PS-LPFGs arc-induced in the SMF-28 and B/Ge#1 fibers. The average values of the temperature sensitivity are also indicated.

**Figure 15 sensors-22-07119-f015:**
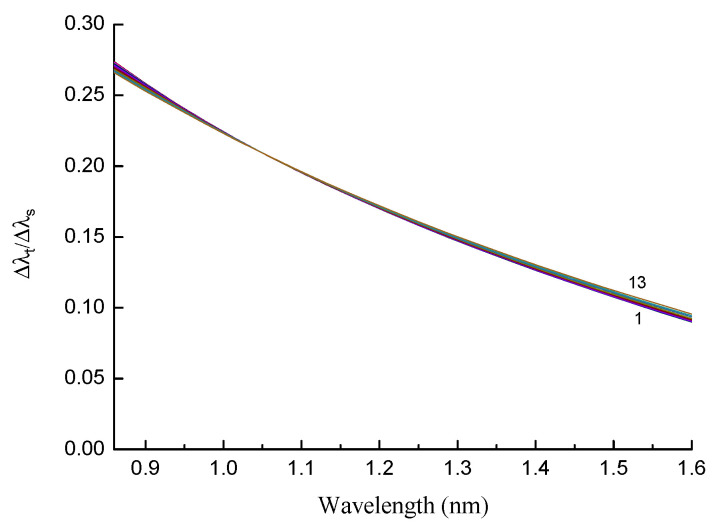
Temperature dependence of figure of merit for a 20 mm grating in B/Ge#1 fiber for modes LP_11_–LP_1,13_ and ΔT=10 K (290–300 K).

**Table 1 sensors-22-07119-t001:** Fabrication parameters of the LPFGs from 2015.

LPFGs 2015	Λ/μm	I/mA	N	Δt/s	w/g
SMF-28	400–600	9	44	1	5
B/Ge #1	540	9	26	0.5	23

**Table 2 sensors-22-07119-t002:** Fabrication parameters of the LPFGs from 2016.

LPFGs 2016	Λ/μm	I/mA	N	Δt/ms	w/g
SMF-28	192–400	11–18	40–400	120–650	2
B/Ge #1	148–192	13.8	140–170	307–320	2
B/Ge #2	182–200	12.7	120–122	660–680	2

**Table 3 sensors-22-07119-t003:** Sellmeier’s coefficients for silica, germanium and boron.

*j*	1	2	3
$D_{S,j}$	0.696750	0.408218	0.890815
$D_{G,j}$	0.8068664	0.7181585	0.8541683
$D_{B,j}$	0.690618	0.401996	0.898817
${D^{\prime }}_{B,j}$	0.6506447	0.3614360	0.9509804
$\lambda_{S,j}$, μm	0.069066	0.115662	9.900559
$\lambda_{G,j}$, μm	0.06897261	0.1539661	11.841931
$\lambda_{B,j}$, μm	0.061900	0.123662	9.098960
${\lambda^{\prime }}_{B,j}$, μm	0.0151863	0.1758124	3.8734989

**Table 4 sensors-22-07119-t004:** Typical temperature sensitivity values obtained using fiber gratings at low temperatures.

Temp. Range Fiber Gratings	S_T_ (pm/K)			
	20 K	100–200 K	200–300 K	Ref
FBG	~0	2.6	2.6	[16]
ORMOCER-coated FBG	1	5	5	[16]
Metallic-coated FBG	15	50	50	[18]
LPFG B/Ge	-	−398	−398	[22]
LPFG B/Ge	~0	−200	−200	[23]
LPFG B/Ge #1	−170	−850	−390	(this study)
PS-LPFG SMF-28	~0	30	30	[25]
PS-LPFG B/Ge #1	−80	−430	−70	[25]

## Data Availability

The data segments can be obtained by contacting the corresponding author.

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
