# Peer review of "High Sensitivity Cryogenic Temperature Sensors Based on Arc-Induced Long-Period Fiber Gratings"

_sensors, 2022, doi:10.3390/s22197119_

Round 1

Reviewer 1 Report

The article named “High Sensitivity Cryogenic Temperature Sensors Based on Arc- Induced Long Period Fiber Gratings” presented a simulation work to investigate the temperature dependencies of the dispersion curve to identify the grating period at cryogenic temperature for the purpose of increment in sensitivity of temperatures. The long-period fiber gratings (LPFG) included SMF28 Corning fiber and two B/Ge co-doped Fiber core fibers. Sensitivity and figures of merit were counted and presented as essential elements for the design of the temperature sensor.  The paper overall carried out great significance in designing the temperature sensor.

1. The schematic diagram of the simulation should be added for a more interactive and detailed approach. 

2. I would suggest representing the data in the tabular form discussed in the “Long-period fiber grating structure” section while discussing the gratings of 2015. 

3. I can see that some of the variables used in the equations are not explained. 

4. I would suggest that the comparison of this study with the previous done should be given in the presentable tabular form. 

5. There are many grammatical errors that should be addressed. 

6. An introductory figure should be added to make the paper more appealing as I have seen only graphical results of the paper. 

7. The previous study related to this study has been discussed in the paper, it will be better to summarize the previous study results in one figure for easier understanding and it will add value to the paper.

8. why did fiber become the two-mode waveguide? considering LP1,1 as the special case. 

9. All the images are graphs so they must have equal sizes and should be aligned either in the center or on the left. 

10. The future work based on this study along with applications should be discussed separately.

Reviewer 2 Report

Using Arc- 2 induced long period fiber gratings, the authors developed high-sensitivity cryogenic temperature sensors. He investigated the evolution of dispersion curves of long-period 13 fiber gratings (LPFGs) from room temperature to 0 K. Gratings arc-in-14 induced in SMF28 fiber and co-doped B/Ge fibers were considered. Prior to publishing, I have major concerns

1. I don't think the introduction is sufficient. In order to understand the different types of LPG-based sensors, he needs to discuss them in more detail. Here are a few references I have found

(a) Satyendra K.Mishra and Kin Seng Chiang, Phenolic-compounds sensor based on immobilization of tyrosinase in polyacrylamide gel on long-period fiber grating, Optics and laser technology, 2020, 131, 106464.

(b) Deependra Tyagi, Satyendra Kumar Mishra, Bing Zou, Congcong Lin, Ting Hao, Ge Zhang, Aiping Lu, Kin Seng Chiang, Zhijun Yang, Nano-functionalized long-period fiber grating probe for disease-specific protein detection, Journal of materials chemistry B, 2018, 6, 386-392.

(c) Satyendra Kumar Mishra, Bing Zou, K.S. Chiang, Wide-Range pH Sensor Based on a SmartHydrogel-Coated Long-Period Fiber Grating, Journal of selected topics in quantum electronics, 2017, 12, 4561.

2. There are a lot of typos in the figures, in many figures he mentions units in nm and in many figures um. In some places, he wrote only grating, and in others, he needs to correct the grating period.

3. The proposed idea needs to be tested in terms of methodology.

4. In the discussion part, he needs to examine the literature and compare his proposed results with other results. In the discussion part, he needs to create a comparison table.

5. The correct unit of temperature is needed. Most of the time, he only wrote K, which is quite confusing. It should be in degrees, Kelvin.

6. English needs to be improved, he has many grammatical errors, and he needs to work on them slightly.

7. It would be helpful if he could give some examples of how to fabricate small grating periods on fiber using different methodologies.

8. His calculation of the figure of merit needs to be clarified by adding some transmission spectra (figure of merit).

9. He needs to change the figure captions because they are inconsistent.

Round 2

Reviewer 1 Report

should be accepted.

Reviewer 2 Report

No Comments